# A Systematic Review of Parkinson’s Disease Pharmacogenomics: Is There Time for Translation into the Clinics?

**DOI:** 10.3390/ijms22137213

**Published:** 2021-07-05

**Authors:** Vladimira Vuletić, Valentino Rački, Eliša Papić, Borut Peterlin

**Affiliations:** 1Clinic of Neurology, Clinical Hospital Center Rijeka, 51000 Rijeka, Croatia; valentino.racki@uniri.hr (V.R.); elisa.papic@uniri.hr (E.P.); 2Department of Neurology, Faculty of Medicine, University of Rijeka, 51000 Rijeka, Croatia; 3Clinical Institute of Medical Genetics, University Medical Center Ljubljana, 1000 Ljubljana, Slovenia; borut.peterlin@kclj.si

**Keywords:** Parkinson’s disease, pharmacogenomics, drug response, levodopa, clinical implementation, personalized medicine

## Abstract

Background: Parkinson’s disease (PD) is the second most frequent neurodegenerative disease, which creates a significant public health burden. There is a challenge for the optimization of therapies since patients not only respond differently to current treatment options but also develop different side effects to the treatment. Genetic variability in the human genome can serve as a biomarker for the metabolism, availability of drugs and stratification of patients for suitable therapies. The goal of this systematic review is to assess the current evidence for the clinical translation of pharmacogenomics in the personalization of treatment for Parkinson’s disease. Methods: We performed a systematic search of Medline database for publications covering the topic of pharmacogenomics and genotype specific mutations in Parkinson’s disease treatment, along with a manual search, and finally included a total of 116 publications in the review. Results: We analyzed 75 studies and 41 reviews published up to December of 2020. Most research is focused on levodopa pharmacogenomic properties and catechol-O-methyltransferase (COMT) enzymatic pathway polymorphisms, which have potential for clinical implementation due to changes in treatment response and side-effects. Likewise, there is some consistent evidence in the heritability of impulse control disorder via Opioid Receptor Kappa 1 (OPRK1), 5-Hydroxytryptamine Receptor 2A (HTR2a) and Dopa decarboxylase (DDC) genotypes, and hyperhomocysteinemia via the Methylenetetrahydrofolate reductase (MTHFR) gene. On the other hand, many available studies vary in design and methodology and lack in sample size, leading to inconsistent findings. Conclusions: This systematic review demonstrated that the evidence for implementation of pharmacogenomics in clinical practice is still lacking and that further research needs to be done to enable a more personalized approach to therapy for each patient.

## 1. Introduction

Parkinson’s disease (PD) is the second most common neurodegenerative disease present today. The incidence and prevalence are highest in the population aged ≥65 years old, making the disease a significant public health burden in the elderly [1]. The clinical course of the disease is progressive and is defined by motor symptoms such as resting tremor, bradykinesia and rigidity, along with a wide variety of non-motor symptoms such as autonomic dysfunction, sleep disorders, cognitive deficits and behavioural changes [2]. The first symptoms appear several years before the classic motor symptoms during the prodromal PD, which is marked by non-specific symptoms like constipation and insomnia [3]. Our understanding of underlying mechanisms in PD has significantly increased over recent years. The main postulated pathological mechanisms in PD include the intracellular aggregation of α-synuclein, which form Lewy bodies [4], as well as the loss of dopaminergic neurons, which first happens in the substantia nigra but later becomes more widespread as the disease progresses [5]. The landmark paper published by Braak et al. describes a gradually evolving pathological severity, starting from the lower brainstem, with a progression to the limbic and neocortical brain regions in the later stages of PD [6].

The variation of clinical states between patients can be significant, even though the underlying mechanisms are similar. Efforts have been made to categorize the disease into varying subtypes. Seyed-Mohammad et al. [7] propose three subtypes based predominantly on clinical characteristics: the mild motor, intermediate and diffuse malignant subtypes. Importantly, findings from the study indicated that neuroimaging correlated better with the subtypes than genetic information, even after incorporating a single “genetic risk score” that encompassed 30 specific PD-related mutations. However, this could also be a consequence of a lack of patients with particular variations in the population they studied [7]. The need to categorize the disease comes from its variability in presentation, response to treatment and incidence of side-effects.

Current treatment options for PD are plentiful, at least in comparison to other neurodegenerative diseases, and offer PD patients extended control of symptom severity as well as an improved quality of life. Unfortunately, no treatment halts the pathological mechanisms that drive disease progression, with most treatment being focused on replacing or enhancing dopamine availability. The golden standard in pharmacologic therapy is dopamine replacement therapy, mainly levodopa, used in synergy with dopamine receptor agonists, monoamine oxidase (MAO) inhibitors or catechol-O-methyltransferase (COMT) inhibitors [8]. The challenge that stems from this type of therapy is the delicate balance between the beneficial and harmful effects that can arise [9]. There is a significant variation in therapy response and side-effect incidence in treating PD, which can be linked to the varied subtypes mentioned earlier, along with increasing evidence of complex environmental and genetic factor interaction [10,11,12]. The consequence of this is the need to fine-tune and personalize the therapy to each patient to account for the variability in drug response [8]. As most treatment is focused on L-dopa, understanding the key players in its metabolism has put the research focus in pharmacogenomics on genes that influence the enzymes and receptors in this pathway [13]. The general principles and goals of pharmacogenomics are to identify the genetic factors behind the varied drug response in individuals, thereby predicting response and paving the way for personalised medicine [14]. The two main areas where the variability of drug response is studied are known as pharmacokinetics and pharmacodynamics. Pharmacokinetics incorporates all processes that affect drug absorption, distribution and metabolism in the body as well as its excretion, while pharmacodynamics focuses on the target actions of the drug. Current evidence suggests that genetic variability and its effects on drug characteristics are concentrated in three major steps: the initial pharmacokinetic processes that ultimately affect the plasma concentration, the capability of drugs in passing the blood-brain barrier (BBB) and finally, the modification of target pharmacodynamic properties of the drug [13]. Expanding the knowledge of the variations that affect these three factors will pave the way for predicting drug response, thus furthering the benefit of a personalized medicine approach in all diseases. Unfortunately, there are currently no clinical guidelines regarding the use of pharmacogenomics in the clinical practice of treating PD, with sparse clinical annotations on relevant databases [15]. Therefore, our aim is to assess the current state of knowledge in this field and the possibility of translation into the clinics.

## 2. Results

Current treatment in PD is focused on alleviating the symptoms and does little to slow down the pathophysiological progression of the disease. As such, the therapy goal is to increase the amount of dopamine to compensate for the loss of dopaminergic neurons. The therapeutic of choice for this is levodopa (L-dopa), which relieves the motor symptoms by increasing the availability of dopamine in the central nervous system (CNS) [16]. All the current pharmaceutical treatment options centre around the dopamine metabolic pathway, which encompasses many genetic pathways. However, there are specific pharmacogenomic properties for different treatment options, as well as differences in pharmacogenomic properties in genotype driven PD.

### 2.1. Drug Specific Pharmacogenomic Properties 

#### 2.1.1. Pharmacogenomics of the Therapeutic Response to L-dopa

Clinically, L-dopa is always combined with dopa decarboxylase (DDC) inhibitors, which causes a switch in L-dopa metabolism to the COMT pathway, thereby increasing the bioavailability of L-dopa in the CNS [16]. The genetic variability of several genes has been implicated in the varied response to L-dopa. *COMT* gene is a protein-coding gene that provides instructions for creating the COMT enzyme, and its polymorphisms are involved in the varied response to numerous CNS diseases and treatments [17,18]. The most studied polymorphism of the *COMT* gene is rs4680 (G>A), which results in a valine to methionine substitution at codon 158 (Val158Met). Single nucleotide polymorphisms of the *COMT* gene form haplotypes that result in lower (A_C_C_G), medium (A_T_C_A) and higher (G_C_G_G) enzyme activity, which, in the case of higher activity, had an impact on the required dosage compared to noncarriers [19] (Table 1). Studies have shown that the higher dosage is required during chronic administration in patients with greater COMT activity, while acute L-dopa administration was unchanged [20,21,22]. Similar changes were observed in a recent study by Sampaio et al., where higher COMT enzyme activity was linked to higher doses of L-dopa required, while no significant changes in dosage were found in lower COMT enzymatic activity compared to the control [23]. Common characteristics of patients that required the higher L-dopa dosage in multiple studies were advanced PD and earlier onset. A contradicting result was published in patients of Korean origins, with no significant association between the rs4680 polymorphism and the response to L-dopa; however, the study population did not have a considerable number of patients with advanced PD [24]. Higher L-dopa doses were needed for patients with *Solute Carrier Family 22 Member 1 (SLC22A1)* gene rs622342A>C polymorphism that encodes the Organic Cation Transporter 1, along with the patients having higher mortality than the control population [25]. On the other hand, lower required doses of L-dopa were found in patients with *Synaptic vesicle glycoprotein 2C (SV2C)* rs30196 polymorphism, as well as in *Solute Carrier Family 6 Member 3 (SLC6A3)* polymorphism after multivariate analysis [26].

#### 2.1.2. Pharmacogenomics of the Side-Effects to L-dopa

Increased incidence of adverse events in L-dopa treatment has been linked with various gene polymorphisms. Although the variations in COMT enzymatic activity on the onset of adverse events is still under debate, several studies have linked the lower COMT enzymatic activity to the increased incidence of motor complications such as dyskinesia, especially in advanced PD [23,27]. Hypothetically, more moderate COMT enzymatic activity could lead to inadequate dopamine inactivation and the accumulation of dopamine in the synaptic cleft, thereby causing the dyskinesias. The same result was not replicated in studies by Watanabe et al. [28] and Contin et al. [22].

There is some evidence that the activation of the Mechanistic target of rapamycin (mTOR) signaling pathway contributes to L-dopa induced dyskinesia. It was indicative of earlier animal studies that the inhibition of mTOR pathways reduces the L-dopa related dyskinesia, most likely due to impaired metabolic homeostasis [78]. These findings were corroborated in a recent human study, by Martin-Flores et al., that found significant associations with several SNPs affecting the mTOR pathway, indicating that the mTOR pathway contributes genetically to L-dopa induced dyskinesia susceptibility [29]. Similarly, a functional *Brain derived neurotrophic factor (BDNF)* Val66Met polymorphism can lead to aberrant synaptic plasticity, which has been associated with L-dopa induced dyskinesia in a single study by Foltynie et al. [30]. Limited evidence has been found in favour of a protective function of the *Dopamine receptor 1 (DRD1)* (rs4532) SNP, shown in a single study by Dos Santos et al. [31]. The effect of *Dopamine receptor 2 (DRD2)* SNP’s on dyskinesia is a point of contention in current literature, as some studies indicate an increased risk of developing dyskinesia [31,32,33], while others revealed a protective effect on the incidence of dyskinesia [34,35]. Interestingly, both studies that show reduced dyskinesias were conducted in the Italian population with the polymorphism DRD2 CAn-STR. Increased risk for developing L-dopa induced dyskinesia was seen in the *Dopamine receptor 3 (DRD3)* rs6280 polymorphism in a Korean population [36]. However, opposing results were found by three research groups, with no evidence of correlation between DRD3 genetic polymorphisms and incidence of dyskinesias [37,38,39]. Lower risk of L-dopa-associated dyskinesias was found in patients with *Homer protein homolog 1 (HOMER1*) rs4704560 G allele polymorphism [40]. Finally, incidence of L-dopa induced dyskinesias was studied for the *dopamine transporter gene (DAT)*, where the presence of two genotypes 10R/10R (rs28363170) and A carrier (rs393795) led to a reduced risk of dyskinesias in an Italian population [41].

Hyperhomocysteinemia is a known complication of L-dopa treatment in PD. The potential dangers of elevated plasma homocysteine are systemic, and include cardiovascular risk, increased risk for dementia and impaired bone health [79]. A SNP C667T (rs1801133) in the *MTHFR* gene is consistently being linked to hyperhomocysteinemia due to L-dopa treatment in several studies. The result of this mutation is a temperature-labile MTHFR enzyme, which ultimately leads to hyperhomocysteinemia [42]. In addition, a study by Gorgone et al. showed that elevated homocysteine levels lead to systemic oxidative stress in patients with this polymorphism [43]. A recent study by Yuan et al. further adds to the claim that homocysteine levels are affected by L-dopa administration, especially in 677C/T and T/T genotypes [44]. A possible option for homocysteine level reduction and alleviation of systemic oxidative stress is the addition of COMT inhibitors to the therapy, which presents a clear possibility for translation of this knowledge into the treatment of patients [79].

There is contradicting evidence regarding whether COMT polymorphisms can influence the incidence of daytime sleepiness in PD patients, with differing results of the pilot and follow-up studies conducted by the same authors [45,46]. Two additional studies by the same primary author revealed an association between sudden-sleep onset and the polymorphisms in hypocretin and DRD2, which was unrelated to a specific drug [47,48]. Furthermore, increased risk of sleep attacks was found in *Dopamine receptor 4 (DRD4)* 48-bp VNTR polymorphism in a German population [39]. The L-dopa adverse effects affecting emetic activity are not uncommon in PD treatment. *DRD2* and *DRD3* polymorphisms both showed an association with an increased risk of developing gastrointestinal adverse effects that do not respond well to therapy in a Brazilian population [32,49]. However, that has not been reproduced in a recent study in a Slovenian population by Redenšek et al. [50].

Mental and cognitive adverse effects of L-dopa are common due to the shared physiological dopaminergic pathways. A significant interaction was found between L-dopa and the *COMT* gene polymorphism in causing a detrimental effect on the activity in task-specific regions of the pre-frontal cortex due to altered availability of dopamine [51,52]. Interestingly, carriers of at least one *COMT* rs165815 C allele had a decreased risk of developing visual hallucinations [50]. In the same study carriers of the *DRD3* rs6280 C allele had higher odds of developing visual hallucinations [50], which is in line with a previous study published by Goetz et al. [53]. Increased risk of developing hallucinations is seen in patients with polymorphisms in the *DRD2* gene [54], *cholecystokinin* gene [55] and *HOMER1* rs4704559 A allele [56], which encodes a protein that possesses a vital function for synaptic plasticity and glutamate signaling. On the other hand, the *HOMER 1* rs4704559 G allele appears to decrease the risk of visual hallucinations [40]. Furthermore, several studies link *BDN*F Val66Met polymorphism to impaired cognitive functioning in PD, but it appears to be irrespective of dopamine replacement therapy and is a genotype-specific trait [57].

Impulse control disorder (ICD) is a well-known complication that can occur in some PD patients after initiating dopamine replacement therapy by either L-dopa or dopamine agonists [58]. Heritability of ICD in a cohort of PD patients has been estimated at 57%, particularly for *Opioid Receptor Kappa 1 (OPRK1)*, *5-Hydroxytryptamine Receptor 2A (HTR2a) and Dopa decarboxylase (DDC)* genotypes [80]. A recent study found a suggestive association for developing ICD in variants of the opioid receptor gene *OPRM1* and the *DAT* gene [59]. Furthermore, there is evidence that polymorphisms in *DRD1* (rs4857798, rs4532, rs265981), *DRD2/ANKK1* (rs1800497) and *glutamate ionotropic receptor NMDA type subunit 2B (GRIN2B)* (rs7301328) bear an increased risk of developing ICD [60,81]. The *DRD3* (rs6280) mutation has also been linked with increased incidence of ICD with L-dopa therapy in studies by Lee et al. [61] and Castro-Martinez et al. [62]. On the other hand, there was no significant association found in *COMT* Val158Met and *DRD2* Taq1A polymorphisms [81]. Even though current data suggest high heritability for developing ICD after initiating dopamine replacement therapy, it should be noted that the effects of individual genes are small, and the development is most likely multigenic.

#### 2.1.3. Dopamine Receptor Agonists

Dopamine receptor agonists (DAs) are often the first therapies initiated in PD patients and are the main alternative to L-dopa [82]. The effectiveness of DAs is lower than L-dopa, and most patients discontinue treatment within three years. Some significance has been found in polymorphisms of the *DRD2* and *DRD3* genes that could influence drug effectiveness and tolerability. A retrospective study by Arbouw et al. revealed that a *DRD2* (CA)n-repeat polymorphism is linked with a decreased discontinuation of non-ergoline DA treatment, although the sample size in this study was small [63]. A pilot study that included Chinese PD patients revealed that the *DRD3* Ser9Gly (rs6280) polymorphism is associated with a varied response to pramipexole [64], which has since been confirmed in a recent study by Xu et al. [65].

Interestingly, the same polymorphism has also been linked with depression severity in PD, indicating that in *DRD3* Ser9Gly patients with Ser/Gly and Gly/Gly genotypes more care should be given to adjusting therapy and caring for non-motor complications [66]. Furthermore, there is evidence from the aforementioned studies that *DRD2* Taq1A polymorphism does not play a significant role in response to DA treatment [64,65,67]. On the other hand, certain Taq1A polymorphisms (rs1800497) have been associated with differences in critical cognitive control processes depending on allele expression [67]. As mentioned earlier, another crucial pharmacogenomic characteristic of DA to bear in mind when administering therapy is the possibility of genotype driven impulse control disorders, which is a problem, especially in *de-novo* PD patients starting DA therapy [80]. Genetic model of polymorphisms in *DRD1* (rs5326), *OPRK1* (rs702764), *OPRM1* (rs677830) and *COMT* (rs4646318) genes had a high prediction of ICD in patients of DA therapy (AUC of 0.70 (95% CI: 0.61–0.79) [68].

#### 2.1.4. COMT Inhibitors

COMT inhibitors are potent drugs that increase the bioavailability of L-dopa by stopping the physiological O-methylation of levodopa to its metabolite 3-O-methyldopa, and can work in tandem with DDC inhibitors [69]. Similar to L-dopa, the presence of the previously mentioned rs4608 *COMT* gene polymorphism modified the motor response to COMT inhibitors entacapone in a small-sample study [83]. Patients with higher COMT enzyme activity had greater response compared to patients with lower COMT enzyme activity during the acute challenge with entacapone [83]. Subsequent studies have not found clinically significance in repeated administration of either entacapone [70] or tolcapone [71], with the impact on opicapone still unknown, meriting further study. Increased doses of carbidopa combined with levodopa and entacapone can improve “off” times, which was shown in a recent randomized trial by Trenkwalder et al., with an even more pronounced effect in patients that had higher COMT enzymatic activity due to COMT gene polymorphisms [72].

Pharmacokinetic studies have shown that COMT inhibitors are metabolized in the liver by glucuronidation, in particular by UDP-glucuronyltransferase UGT1A and UGT1A9 enzymes [82]. Hepatotoxicity is a known rare side-effect of tolcapone [73], with only sparse reports of entacapone hepatotoxicity [84]. Several studies indicate that SNPs in the UGT1A and UGT1A9 are responsible for these adverse events, which can cause inadequate metabolism and subsequent damage to the liver by the drugs [74,75,85,86]. Interestingly, opicapone has not demonstrated evident hepatotoxicity related adverse events, while in-vitro studies show a favorable effect on hepatocytes when compared to entacapone and tolcapone [76].

#### 2.1.5. MAO Inhibitors

MAO inhibitors are used with L-dopa to extend its duration due to reduced degradation in the CNS. Most MAO inhibitors used today in PD treatment (e.g., selegiline, rasagiline) are focused on blocking the MAO-B enzyme that is the main isoform responsible for the degradation of dopamine [87]. There have not been many studies performed to assess MAO inhibitor pharmacogenetic properties. Early clinical studies with rasagiline did reveal an inter-individual variation in the quality of response that could not be adequately explained at that time [88]. Masellisi et al. conducted an extensive study using the ADAGIO study data to identify possible genetic determinants that can alter the response to rasagiline. They identified two SNPs on the *DRD2* gene that were associated with statistically significant improvement of both motor and mental functions after 12 weeks of treatment [89].

### 2.2. Genotype Specific Treatment and Pharmacogenomic Properties 

Gene variations that influence pharmacogenomic properties and treatment in PD are not only focused on the metabolic and activity pathways of the drugs. There is a wide number of genes that are linked to monogenic PD, but only some had their association proven continuously in various research studies. Mutations in the genes coding *α-Synuclein (SNCA)*, *Leucine-rich repeat kinase 2 (LRRK2)*, *vacuolar protein sorting-associated protein 35 (VPS35)*, *parkin RBR E3 ubiquitin-protein ligase (PRKN)*, *PTEN-induced putative kinase 1 (PINK1)*, *glucocerebrosidase (GBA)* and *oncogene DJ-1* [77] have mostly been found before the onset of genome-wide association studies, while many candidate genes found after are yet to be definitively proven to cause a significant risk for PD. Importantly, the currently known candidate genes can explain only a small fraction of cases where there is a known higher familial incidence of PD [90]. It is remarkable, however, that assessing polygenic risk scores and combining those with specific clinical parameters can yield impressive sensitivity of 83.4% and specificity of 90% [91] (Table 2.). The unfortunate consequence of the rapid expansion of knowledge in the field and amount of target genes is that the studies assessing pharmacogenomics of these gene variants are not keeping up.

#### 2.2.1. LRRK2

Current evidence, albeit limited, points to differences in treatment response between various genotypes of monogenic PD. Mutations in the *LRRK2* gene are known to cause familial PD, especially in North African and Ashkenazi Jew populations [92]. *LRRK2* protein has a variety of physiological functions in intracellular trafficking and cytoskeleton dynamics, along with a substantial role in the cells of innate immunity. It is yet unclear how mutations in *LRRK2* influence the pathogenesis of PD, but there is numerous evidence that links it to a disorder in cellular homeostasis and subsequent α-synuclein aggregation [105]. Results in in vitro and in vivo animal model studies for inhibition of mutant *LRRK2* are promising, and in most cases, confirm a reduced degeneration of dopaminergic neurons [106]. The biggest challenge of human trials has been creating an LRRK2 inhibitor that can pass the blood-brain barrier, which was overcome by Denali Therapeutics, and the phase-1b trial for their novel LRRK2 inhibitor has been completed and is awaiting official results [105]. Furthermore, *LRRK2*-associated PD has a similar response to L-dopa compared to sporadic PD, with conflicting results for the possible earlier development of motor symptoms [13]. Pharmacogenomics in *LRRK2* associated PD are linked to specific genotype variants. G2019S and G2385R variants in *LRRK2* have been linked as predictors of motor complications due to L-dopa treatment, along with requiring higher doses during treatment [107]. On the other hand, G2019S carrier status did not influence the prevalence of L-dopa induced dyskinesias in a study by Yahalom et al. [93]. Furthermore, a study covering the pharmacogenetics of Atremorine, a novel bioproduct with neuroprotective effects of dopaminergic neurons, found that *LRRK2* associated PD patients had a more robust response to the compound, along with several genes that cover metabolic and detoxification pathways [94].

#### 2.2.2. SNCA

*SNCA* gene encodes the protein α-synuclein, now considered a central player in the pathogenesis of PD due to its aggregation into Lewy-bodies. SNP’s in the *SNC*A gene are consistently linked to an increased risk of developing PD in GWAS studies in both familial and even sporadic PD [95]. In cases of autosomal dominant mutations, there is a solid L-dopa and classical PD treatment response, albeit with early cognitive and mental problems, akin to GBA mutations [108]. There are several planned therapeutic approaches suited for *SNCA* polymorphism genotypes which include: targeted monoclonal antibody immunotherapy of α-synuclein [96], downregulation of *SNCA* expression by targeted DNA editing [109] and RNA interference of *SNCA* [97]. Roche Pharmaceuticals has developed an anti-α-synuclein monoclonal antibody which is in a currently ongoing phase two of clinical trials [110]. Two other methods are still in preclinical testing, and their development shows promise for the future.

#### 2.2.3. GBA

Glucocerebrosidase mutations represent a known risk factor for developing PD. *GBA* mutation associated PD is characterized by the earlier onset of the disease, followed by a more pronounced cognitive deficit and a significantly higher risk of dementia [98]. Gaucher’s disease (GD) is an autosomal recessive genetic disorder that also arises from mutations in the *GBA* gene. The current enzyme replacement and chaperone treatment options for systemic manifestations of GD are not effective enough in treating the neurological manifestations of the disease as they are not able to reach the CNS [111]. Three genotype-specific therapies to address the cognitive decline are currently being tested with promising early results, with two focusing on the chaperones ambroxol [112] and LTI-291 to increase glucocerebrosidase activity and the third focusing on reducing the levels of glucocerebrosidase with ibiglustat [98]. There is growing evidence that GBA associated PD is often marked by rapid progression with many hallmarks of advanced PD, such as higher L-dopa daily dose required to control motor symptoms [99]. However, current research does not show a significant influence of *GBA* mutations on L-dopa response properties with adequate motor symptom control [100]. A single study by Lesage et al. in a population of European origin linked a higher incidence of L-dopa induced dyskinesias in GBA-PD patients [113], but that has not been replicated in a more recent study by Zhang et al. in a population of Chinese origin [101].

#### 2.2.4. PRKN/PINK1/DJ1

Mutations in the *PRKN* gene can lead to early onset PD, characterized by a clinically typical form of PD that is often associated with dystonia and dyskinesia [102]. Patients with *PRKN* mutations generally have excellent and sustained responses to L-dopa, even in lower doses than in sporadic PD [103]. Dyskinesias can occur early on in the course of the disease with very low doses of L-dopa [114], while dystonia in these patients was not found to be linked to L-dopa treatment [102]. Furthermore, patients with *PINK1* mutations have a similar disease course as PRKN mutation carriers, with a good response to L-dopa treatment, but early dystonia and L-dopa induced dyskinesias [102]. Pharmacogenomic properties and genotype-specific treatment of several other gene mutations in PD such as *VPS35* and *DJ1* have not yet been characterized fully due to the rarity of cases and are currently a focus of several studies that as of writing do not have preliminary results available [90,115,116].

## 3. Discussion

There has been considerable progress in the field of pharmacogenomics in Parkinson’s disease. The main question in the field is whether we can use the current knowledge in clinical practice to benefit the patients. The data on Parkinson’s disease in PharmGKB, a pharmacogenomics database, are sparse, with only ten clinical annotations with most being supported by a rather low level of evidence, which is clear from this systematic review as well [15]. Most of the pharmacogenomic studies that focus on antiparkinsonian drugs are highly centered on L-dopa and its metabolism. The current evidence on the pharmacogenomics of therapeutic response to L-dopa is contradictory, with most studies focusing on the COMT gene polymorphisms. The differences between studies limit the potential for clinical use. However, there is potential to clarify the effects of *COMT* gene polymorphisms by further studies analyzing the enzymatic activity in various genotypes and the L-dopa dosage and therapeutic response. More robust evidence is present for the pharmacogenomics of side-effects in L-dopa or dopaminergic therapy. The most studied motor complication of L-dopa therapy is treatment-induced dyskinesias. Looking at the evidence, we can see that there are numerous reports focusing on various genes, although often with contradictory results in *COMT*, *DRD2* and *DRD3* genes. On the other hand, SNPs in the *mTOR* pathway genes, *BDNF*, *HOMER1* and *DAT* have been implicated in either increased or reduced risk for dyskinesias, but with single studies that are yet to be corroborated in larger cohorts. Other side-effects such as cognitive decline, visual hallucinations and daytime sleepiness have been implicated in various polymorphisms of the *COMT, DRD2, DRD3, HOMER1* and *BDNF* genes, but lack consistency in the results to consider current clinical implementations.

Hyperhomocysteinemia and ICD are known complications of dopaminergic therapy, and both have been consistently linked with genetic factors [42,43,79]. Specifically, mutations in the *MTHFR* gene can increase the incidence of hyperhomocysteinemia, which could be ameliorated by the addition of COMT inhibitors to therapy, presenting a possibility for clinical interventions based on pharmacogenomic testing. The same can be said about ICD, where genetic models are gaining accuracy with each new study in the field [60,62]. Potential for clinical use can especially be seen in younger patients which are only starting dopamine agonist therapy, as polymorphisms in DRD1 (rs5326), *OPRK1* (rs702764), *OPRM1* (rs677830) and *COMT* (rs4646318) genes showed a high prediction rate of ICD [68]. There is evidence that polymorphisms in *DRD2* and *DRD3* gene could also cause these side-effects, leading to earlier discontinuation of DA therapy in patients. There is clear potential for clinical implementation in this area, and future goal should be to establish studies with larger cohorts in order to improve the genetic prediction models.

There is lacking evidence regarding the pharmacogenomic properties of other drugs used in PD, such as COMT and MAO inhibitors. However, there is some evidence that mutations leading to varied COMT enzyme activity could have an influence on the potency of COMT inhibitors, but the results are not consistent [70,71]. More consistent results have been found regarding entacapone hepatotoxicity, with several studies indicating that SNP’s in the *UGT1A* and *UGT1A9* could lead to this adverse effect [74,75,76]. MAO inhibitors are known to have inter-individual variation, which is still not explained in current studies, with a single study reporting improved motor and mental functions in *DRD2* gene SNP. Taken together, the pharmacogenomic data regarding COMT and MAO inhibitors are still not strong enough to make any recommendations for clinical implementation.

Finally, pharmacogenomics in PD also encompasses changes that occur in specific differences in genotype-associated PD. Three of the most studied single gene mutations are the *LRRK2, GBA* and *SNCA* gene mutations. Published studies covering L-dopa treatment with these mutations have contradicting results depending on the populations studied, which makes it difficult to give any firm recommendations regarding treatment optimization [93,94,96,101,102]. The current evidence for *PRKN, PINK1* and *DJ1* point to a sustained L-dopa response with lower doses, albeit with early motor complications that include dyskinesias and dystonia [103,114,115]. Therefore, this clinical phenotype can raise suspicions of these mutations and lead to earlier genetic testing and treatment optimization. However, the number of cases analyzed is low due to the rarity of these mutations, and further studies are required to confirm these early findings.

## 4. Materials and Methods

We have done a systematic search of articles indexed in Medline and Embase from its inception to July of 2020 focused on the pharmacogenomics in Parkinson’s disease using a strategy similar to what was described by Corvol et al. [13]. The search terms included: Genetic Variation (MeSH), Genotype (MeSH), Genes (MeSH), Polymorphism, Allele, Mutation, Treatment outcome (MeSH), Therapeutics (MeSH), Pharmacogenomic (MeSH), Pharmacogenetics (MeSH), Adverse effects (MeSH Subheading), Toxicogenetics (MeSH) and Parkinson’s disease (MeSH). The articles included in the search were clinical trials, meta-analysis, and randomized controlled trial, with excluding case reports and reviews, with additional filters of human studies and English language. We included studies that had a clear methodology regarding study population and main findings. Exclusion criteria were articles not written in English, lacking study population information and findings not relevant to the theme of pharmacogenomics in PD. Several reviews were added into the overall analyzed papers using manual searches through websites and citation searching. PharmGKB database was accessed as well using the search parameter “Parkinson’s disease” to view current clinical annotations present for PD [15].

The systematic literature search in Medline and Embase revealed 15,778 potential publications, which were first automatically and then manually filtered to exclude studies that do not fit the inclusion criteria (Figure 1). We included 75 studies, with the final count being 116 after adding publications found through manual search that include reviews covering this topic, along with studies focused on genotype specific PD forms (Figure 1).

## 5. Conclusions

Most pharmacogenomic data for PD treatment present today are still not consistent enough to be entered into clinical practice, and further studies are required to enable a more personalized approach to therapy for each patient. The main findings can be summarized as follows:Most evidence from the analyzed studies is found via secondary endpoints, which limits their power, with small sample size also being a diminishing factor.Conflicting reports between varied populations could be a consequence of low sample sizes and unaccounted interactions, which ultimately leads to low confidence in the data currently available.The most promising avenues for clinical implementation of pharmacogenetics lie in the current findings of impulse control disorders and hyperhomocysteinemia, where the available data are more consistent.Most of the studies focus on L-dopa and DA, and greater focus should also be given to other PD treatment options such as MAO-B and COMT inhibitors.

Even though the wealth of knowledge is rapidly increasing, there are still not enough consistent data to make quality choices in the clinical treatment of patients. Studies that have a clear focus on pharmacogenomic properties of antiparkinsonian drugs are key for consolidating the current information and for the translation into clinical practice.

## Figures and Tables

**Figure 1 ijms-22-07213-f001:**
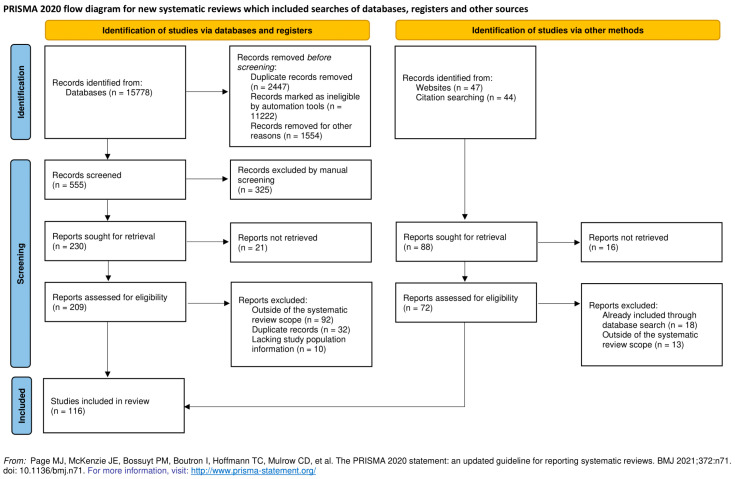
Prisma flow diagram for the systematic review [117].

**Table 1 ijms-22-07213-t001:** Pharmacogenomic studies cited in the systematic review.

References	Study Design	Population (Ethnicity)	Main Finding
Bialecka M et al. [19]	Genetic screening for *COMT* SNP-s and the association of *CO*MT haplotypes with the dose and complications of levodopa therapy in PD patients	679 study participants (322 PD and 357 controls)—participants genotyped for four SNPs in the *COMT* gene	The frequency of G_C_G_G (high activity) haplotype carriers was higher in late onset PD patients (*p* = 0.04) compared with controls
Cheshire et al. [20]	Influence of SNP in *COMT, MAO-A* and *BDNF* on LID	285 Parkinson disease patients	Individual SNPS in *BDNF, COMT* and *MAO-A* required higher doses of levodopa
Bialecka M et al. [21]	Prospective study over 5 years	95 patients with sporadic PD divided into 2 groups (group 1 treated with levodopa < 500 mg/daily; group 2 > 500 mg/daily)	Higher frequency of *COMT* (L/L) homozygotes in the group treated with lower doses of levodopa
Contin M et al. [22]	Prospective study—serial measurement of plasma levodopa, finger-tapping and dyskinesia ratings	104 patients with PD	No clinically relevant levodopa response associated with the COMT polymorphism
Sampaio et al. [23]	Genetic screening for *MAO-B* and *COMT* SNP followed by a multivariate analysis (sex, duration of disease, levodopa treatment duration)	162 Brazillian PD patients treated with levodopa split into 2 groups according to levodopa dose	Patients carrying *MAO-B*(rs1799836) A and AA genotypes and *COMT* (Rs4680) LL genotype suffered more frequently from LID; male population with MAO-B treated with higher doses of levodopa(*p* = 0.04)
Lee et al. [24]	Genetic screening for *COMT* SNP-s in PD, multiple system atrophy and controls; association with response to levodopa	73 Korean patients with PD, 29 with MSA and 49 controls	No significant association between the rs4680 polypmorphism and the response to L-dopa
Becker et al. [25]	Genetic screening for *OCT1* SNP and the correlation with L-dopa dosage	7983 Caucasians aged 55 years and older who had a first prescription for levodopa between July 1st 1991 and January 1st 2008	Higher L-dopa doses were needed for patients with SLC22A1 gene rs622342A>C
Altmann et al. [26]	Multivariate analysis of genetic polymorphisms in relation to L-dopa dosage	224 Parkinson’s disease patients	Lower required doses of L-dopa in patients with *SV2C* rs30196 polymorphism and in *SLC6A3* polymorphism
de Lau LM et al. [27]	Prospective study	219 patients with PD without dyskinesias at baseline	The A-allele of the *COMT* Val158Met polymorphism was related to an increased risk of developing dyskinesias during follow-up
Watanabe et al. [28]	Genetic screening of *COMT* Val-108-Met polymorphism	121 Japanese patients with Parkinson’s disease (PD) and 100 controls.	Patients with homozygosity for the low-activity allele showed a tendency to exhibit the “wearing-off” phenomenon compared with controls
Martin-Flores et al. [29]	Genetic screening for *mTOR* genetic pathway mutations	401 patients with PD of European origin from the northeastern part of the Iberian peninsula	Patients with SNP’s in the mTOR signalling cascade could have increased severity and onset of L-dopa induced dyskinesias. SNP rs1043098 and rs1043098 in the *EIF4EBP2* gene, *RICTOR* rs2043112, and *PRKCA* rs4790904 had increased LID onset. Patients with SNP in *HRAS* rs12628, *PRKN* rs1801582 and also with a four-loci epistatic combination involving *RPS6KB1* rs1292034, *HRAS* rs12628, *RPS6KA2* rs6456121 and *FCHSD1* rs456998 had increased LID severity
Foltynie et al. [30]	Genetic screening for *BDNF* genotypes	315 patients from the UK, unknown origin	Patients with the functional met allele *BDNF* polymorphism is associated with a significantly higher risk of developing dyskinesias
dos Santos et al. [31]	Targeted genotyping for *DRD2/ANKK1* (rs1800497) and *SLC6A3/DAT1* (rs28363170) phenotypes	195 idiopathic Brazilian PD patients	Association between the occurrence of dyskinesia with an increased greater disease severity, higher L-DOPA dose and use of dopamine agonist
Rieck et al. [32]	Targeted genotyping for the variants in the *DRD2/ANKK1* gene region (-141CIns/Del, rs2283265, rs1076560, C957T, TaqIA and rs2734849)	199 Brazilian PD patients	Carriers of the TTCTA haplotype show an increased risk for the presence of dyskinesia (*p* = 0.007; 1.538 [95% CI: 1.126–2.101])
Strong et al. [33]	Targeted genotyping of the mu opioid receptor gene and the *DRD2* gene	92 USA PD patients of unknown origin that had levodopa induced dyskinesias	G-allele of the A118G polymorphism of the mu opioid receptor is associated with an increased risk of earlier dyskinesia onset. Early dyskinesia was linked to the *DRD2* 14 and 14/15 alleles
Zappia et al. [34]	Genotyping analysis of the intronic CA dinucleotide short tandem repeat (CA_n_-STR) polymorphisms in the *DRD2* gene	215 PD patients from southern Italy	Genetic factors related to the *DRD2* CAn-STR polymorphism were not independent predictors for PDD in the total population, but they had a strong protective effect on the appearance of PDD when the multivariate analysis was performed in men. In women, a genetic protective effect on PDD was not evident
Oliveri et al. [35]	Genotyping analysis of polymorphisms in the *DRD1* and *DRD2* genes	136 sporadic PD patients and 224 population control subjects	*DRD1* polymorphisms were not associated with the risk of developing PD or peak-dose dyskinesias. The 15 allele *DRD2* gene polymorphism was more frequent in parkinsonian subjects than in control subjects. Frequency of both the 13 allele and the 14 allele *DRD2* gene plymorphism was higher in non-dyskinetic than in the dyskinetic PD subjects
Lee et al. [36]	Targeted genotyping for six genetic variants (*DRD2* Taq1A (=g.32806C>T, rs1800497), *DRD3* p.S9G (rs6280), *GRIN2B* c.2664C>T (rs1806201), c.366C>G (rs7301328), c.-200T>G (rs1019385) and *5-HTTLPR*)	503 PD patients and 559 healthy controls of Korean origin	*DRD3* p.S9G variant is exclusively associated with diphasic dyskinesia, with the AA genotype likely shortening the durations of the dyskinesias; Peak dose dyskinesias were not associated with any of the six analysed genetic variants
Kaiser et al. [37]	Retrospective noninterventional study focusing on *DRD2, DRD3* and *DRD4* gene polymorphisms.	183 idiopathic German PD patients	The polymorphisms of *DRD2, DRD3,* and *DRD4* were not associated with the risk to develop adverse effects of L -dopa. Patients with psychosis or dyskinesia carried the nine copy allele 40-bp VNTR of the *DAT* more frequently than nonafflicted patients
Wang et al. [38]	Genotype association study of *DRD2* TaqIA, *DRD3* BalI and MspI polymorphisms	140 idiopathic USA PD patients	Findings suggest that *DRD2* TaqIA polymorphism may be associated with an increased risk for developing motor fluctuations in PD
Paus et al. [39]	Database analysis; Association study of *DRD3* Ser9Gly genotype	690 German PD patients	Stepwise regression analysis revealed no effect of *DRD3* Ser9Gly on chorea, dystonia or motor fluctuations in PD. Increased risk of sleep attacks in *DRD4* 48-bp VNTR polymorphism
Schumacher-Schuh et. [40]	Genotyping of the *HOMER1* gene for rs4704559, rs10942891 and rs4704560 polymorphisms	205 Brazilian PD patients	The rs4704559 G allele was associated with a lower prevalence of dyskinesia and visual hallucinations
Purcaro et al. [41]	Targeted genotyping of *DAT* gene polymorphisms (rs28363170, rs393795)	181 Italian PD patients	*DAT* gene 10R/10R (rs28363170) and A carrier (rs393795) of the *DAT* gene reduces the risk of LID occurrence during long-term therapy with l-DOPA (OR = 0.31; 95% CI, 0.09–0.88)
De Bonis ML et al. [42]	Genotyping analysis of A SNP C667T (rs1801133)	44 PD patients treated with L-Dopa (20 with concomitant dopamineagonists, group A) and 12 patients L-Dopa untreated	L-Dopa administration in hyperhomocysteinemic PD patients can lower intracellular concentration of (AdoMet) in erythrocytes (RBC) with hyperhomocysteinaemia causing a significant increase in S-Adenosylhomocysteine (AdoHcy) level; may lead to drug resistance through COMT upregulation
Gorgone et al. [43]	Genotype screening for the methylenetetrahydrofolate reductase (*MTHFR*) gene polymorphism	60 Italian PD patients and 82 healthy subjects	Patients with a TT677 mutated genotype had higher homocysteine and Coenzyme Q10 levels and needed a lower L-dopa daily dose
Yual et al. [44]	Targeted genotyping of the *MTHFR* gene	48 L-dopa treated patients, 28 non-treated PD patients and 110 control of Taiwanese origin	Genetic C677T and A1298C polymorphisms in 5,10-methylenetetrahydrofolate reductase (*MTHFR*) and levodopa therapy in Parkinson’s disease (PD) may increase homocysteine (Hcy) level
Frusher et al. [45]	Genotype screening of D4 receptor of *COMT*(alleles LL,LH,HH) and the correlation with daytime sleepiness	46 PD patients	Higher ESS (Epworth Sleepiness Scale) in LH/LL alleles
Rissling et al. [46]	Genotype screening of *COMT* rs4680 polymorphism and the correlation with daytime sleepiness	240 patients with PD (70 with the met-met (LL), 116 with the met-val (LH), and 54 with the val-val (HH) genotype	No clinical relevance of *COMT* in daytime sleepiness in PD
Rissling et al. [47]	Genotype screening of preprohypocretin polymorphisms and the correlation with sudden onset of sleep	132 PD patients and 132 PD patients without sudden onset of sleep	A significant association between the (-909T/C) preprohypocretin polymorphism and sudden onset of sleep in Parkinson disease
Rissling et al. [48]	Genotype screening of *D2,D3,D4* polymorphisms; association with sudden onset of sleep	137 PD patients with SOS(sudden onset of sleep) and 137 PD patients without SOS	A significant association between the dopamine *D2 receptor gene* polymorphism Taq IA and SOS in PD
Rieck et al. [49]	Genotype association study of *DRD2* and *DRD3* gene polymorphisms and gastrointestinal symptoms induced by levodopa therapy	217 Brazilian PD patients	*DRD2* Ins/Ins and *DRD3* Ser/Ser genotypes were independent and predictors of gastrointestinal symptoms associated with levodopa therapy
Redenšek et al. [50]	Retrospective cohort study	31 unrelated PD patient	carriers of at least one *DRD3* rs6280 C allele and CC homozygotes had higher odds for this adverse event
Nombela et al. [51]	Prospective cohort study focused on cognitive decline and *COMT* Val158Met (rs4680), *MAPT* (rs9468) H1 vs. H2 haplotype and *APOE-**e**2, 3, 4* polymorphisms	168 UK PD patients and 85 matched controls	All three analysed genotypes had significant association with cognitive decline, with associations relating to L-dopa therapy in the COMT gene
Williams-Gray et al. [52]	Genotype associative study on cognitive decline in association to *COMT* val(158)met genotype and dopaminergic medication	29 medicated patients with early PD	Significant underactivation across the frontoparietal attentional network
Goetz et al. [53]	Case control study	44 patients with PD and chronic hallucinations and 44 patients with PD without	Carriers of DRD3 rs6280 C allele may have higher odds of developing visual hallucinations
Makoff et al. [54]	Genotype association study of *DRD2* and *DRD3* polymorphims and drug-induced hallucinations	155 white Caucasian PD patients from the UK	No association was found with the whole group of hallucinating patients and their controls. However, an association was found with late-onset hallucinations and the C allele of the TaqIA polymorphism, 10.5 kb 39 to *DRD2*
Wang et al. [55]	Case control study	160 Chinese patients with Parkinson’s disease and 160 controls	Visual hallucinations in PD are associated with cholecystokinin -45C>T polymorphism; also in the presence of the cholecystokinin-A receptor TC/CC genotype
De Luca et al. [56]	Genotyping association study of *HOMER1* gene and development of psychotic symptoms in PD	131 sporadic PD patients from southern Italy	allele A of the rs4704559 marker linked to increased susceptibility to psychotic symptoms in PD *p* = 0.004
Wang et al. [57]	Meta-analysis focused on the association between *BDNF G196A* (Val66Met) polymorphism and cognitive impairment in PD patients	Six studies involving 532 PD patients and 802 controls	*G196A* (Val66Met) polymorphism is significantly associated with cognitive impairment in PD, especially in Caucasian populations
Gatto et al. [58]	Genotyping multivariate study of SNP-s and Impulse control disorder(ICD)	276 patients with PD	*OPRK1* polymorphism rs702764 significantly predicted incident ICD behaviour; polymorphisms of the *DRD1, DRD2, DRD3, HTR2A* and *GRIN2B* genes also associated with ICD in patients with PD
Cormier-Dequaire et al. [59]	Multicenter case-control genotype association study	172 French Caucasian patients and 132 controls	No variant was significantly associated with impulse control disorders or related behaviors after correction for multiple testing, although the 2 top variants were close to significant (*OPRM1* rs179991, *p* = 0.0013; Bonferroni adjusted *p* = 0.065; *DAT1* 40-base pair variable number tandem repeat; *p* = 0.0021; Bonferroni adjusted *p* = 0.105)
Zainal Abidin et al. [60]	Multivariate association study of SNPs and increased risk of ICD development	52 Malaysian PD patients with 39 without ICB	*DRD1* (rs4532 and rs4867798), *DRD2/ANKK1* rs1800497] and glutamate (*GRIN2B* rs7301328) receptor genes confer increased risk of ICD development among PD patients
Lee et al. [61]	Genotype association study genotypes *DRD3* p.S9G, *DRD2* Taq1A, *GRIN2B* c.366C>G, c.2664C>T and c.-200T>G, and the promoter region of the *serotonin transporter gene (5-HTTLPR)* with the incidence of ICD’s	404 Korean PD patients and 559 healthy controls	Variants of *DRD3* p.S9G and *GRIN2B* c.366C>G may be associated with the appearance of ICD in PD
Castro-Martinez et al. [62]	Genotype association study of rs6280 *DRD3* single nucleotide variation (SNV) (Ser9Gly) and incidence of ICD’s	199 Hispanic PD patients	Behavioral addictions in PD are associated with an early onset of the disease, the rs6280 *DRD3* SNV and the type of dopamine agonist
Arbouw et al. [63]	Genotype association study of pharmacogenetic determinants for the discontinuation of non-ergoline dopamine agonists	90 Dutch PD patients	This study identified apomorphine use and levodopa dosages between 500 and 1000 mg as non-genetic and the 15× *DRD2* CA repeat allele as genetic determinants for the discontinuation of non-ergoline DA treatment in patients with PD
Liu et al. [64]	Genotype association study focusing on the response to pramipexole in PD patients	30 Chinese PD patients	*DRD3* Ser9Gly polymorphisms are significantly associated with the therapeutic efficacy of pramipexole in Chinese patients with PD
Xu et al. [65]	Genotype association study of *DRD2* CA n-STR and *DRD3* Ser9Gly polymorphisms with Parkinson’s disease and response to dopamine agonists	168 PD patients of Chinese origin and 182 controls	Genotype in *DRD3* Ser9Gly was the main factor determining different doses of DAs and PD patients carrying Gly/Gly genotype require higher doses of pramipexole for effective treatment
Zhi et al. [66]	Genotype association study of *DRD3* Ser9Gly polymorphism and depression severity in Parkinson’s disease	61 PD patients of Chinese origin and 47 controls	*D3 gene* Ser9Gly polymorphism might be associated with the severity of depression characterized by anhedonia in PD patients
Paus et al. [67]	Genotype association study of the *DRD2* TaqIA polymorphism and demand of dopaminergic medication in Parkinson’s disease	607 PD German patients of varied origin	*DRD2* TaqIA polymorphism alone has no pivotal role for interindividual variability of dopaminergic requirement in PD
McDonell et al. [68]	Genotype association study of *DRD2* and *DRD3* gene polymorphisms and the incidence of ICD	28 USA PD patients	Patients with the rs1800497 *Taq1A* (A1) polymorphism (A1/A1 or A1/A2: 11 subjects) showed improved proficiency to suppress impulsive actions when on DAAg; conversely, patients with the A2/A2 allele (14 patients) became less proficient at suppressing incorrect response information on DA; Polymorphisms in rs6277 and rs6280 were not associated with a differential medication response
Erga et al. [69]	Whole-exome sequencing study of 17 genes connected to ICD	119 Norwegian PD patients	Eleven SNPs were associated with ICDs, and the four SNPs with the most robust performance significantly increased ICD predictability (AUC = 0.81, 95% CI 0.73–0.90) compared to clinical data alone (DA use and age; AUC = 0.65, 95% CI 0.59–0.78); The strongest predictive factors were rs5326 in *DRD1*, which was associated with increased odds of ICDs, and rs702764 in *OPRK1*, which was associated with decreased odds of ICDs
Corvol et al. [70]	Randomized crossover clinical trial focused on the effect of *COMT* Val158Met polymorphism to the entacapone response in PD patients	58 French PD patients	The *COMT^HH^* genotype in PD patients enhances the effect of entacapone on the pharmacodynamics and pharmacokinetics of levodopa
Lee et al. [71]	Genotype association study of *COMT* genotypes and entacapone efficacy	65 PD patients with entacapone therapy	After entacapone treatment, the mean of the percentage reduction of daily levo-dopa dose for each individual was significant in patients with HH and HL genotype of *COMT*
Chong et al. [72]	Genotype association study of *COMT* genotypes and tolcapone efficacy	24 PD patients who completed tolca-pone clinical trials	no substantial effect of *COMT* genotype relative to clinical response to the COMT inhibitor tolcapone
Trenkwalder et al. [73]	Randomized double-blind crossover multicenter study	117 German PD patients	Patients with high-activity *COMT* genotypes Val/Met and Val/Val had a reduced “off” time
Liu et al. [74]	Genotyping analysis of *UGT1A, UGT2B* SNPs	148 liver samples (125 of European and 23 of African descent)	*UGT* SNP variants contribute to variability in the metabolism of certain drugs and can lead to adverse effects due to inadequate metabolism
Yamanaka et al. [75]	Genotyping analysis of *UGT1A9* SNPs	87 Japanese, 50 Caucasian and 50 African-American participants	The mutant allele with one base insertion in the promoter region of the *UGT1A9* gene would alter the level of enzyme expression and the metabolism of those drugs that are substrates of *UGT1A9*
Ferrari et al. [76]	Genotyping association study of *UGT1A9* SNPs and COMT inhibitor induced toxicity	52 Parkinson’s disease (PD) patients on COMT inhibitors without evidence of adverse reactions and 11 PD patients who had been withdrawn from COMT inhibitors due to adverse reactions	In PD patients *UDP-glucuronosyltransferase 1A9* genotypes are associated with adverse reactions to COMT inhibitors
Masellis et al. [77]	Genotyping association study of *DRD2* SNPs and response to rasagiline	692 available DNA samples from a placebo-controlled clinical trial of the monoamine oxidase B inhibitor	rs2283265 and rs1076560 were found to be significantly associated with a favourable peak response to rasagiline at 12 weeks in early Parkinson’s disease

**Table 2 ijms-22-07213-t002:** Genotype specific Parkinson’s disease studies cited in the manuscript.

References	Study Design	Methodology/Specific Mutation Studied	Main Finding
Nalls et al. [92]	Multicenter population-based modelling study	367 PD patients and 165 controls for the model Model tested on 825 PD patients and 261 controls	The developed model for disease classification could distinguish participants with PD and controls with high sensitivity (0·834, 95% CI 0·711–0·883) and specificity (0·903, 95% CI 0·824–0·946)
Shu et al. [93]	Meta-analysis	66 studies comprising 23,402 PD patients. Association of *LRRK2* with clinical features of PD	Clinical heterogeneity in *LRRK2*-associated PD among different variants, especially for G2019S and G2385R
Yahalom et al. [94]	Genotype association study of G2019S *LRRK2* mutation with regards to L-dopa induced dyskinesias	349 Israeli PD patients (222 Askenazi-Jewish)	The prevalence of LID was non-significantly higher among carriers (22/33, 66.7%) than non-carriers (168/316, 53.2%, *p* = 0.15)
Cacabelos et al. [95]	Multivariate association study of *atremorin* and *LRRK2* variants	183 PD patients split into 2 categories: 135 drug-free patients (DF-PD) who had never before received any anti-parkinsonian medication 48 patients chronically treated with anti-parkinsonian drugs (CT-PD) (>1 y) (n = 48)	*LRRK2* associated PD patients had a more robust response to the compound atremorin
Nishioka et al. [96]	Genome wide association study	103 Japanese patients with autosomal dominant PD [43 male and 60 female with a mean age at onset of 50.9613.9 years (6SD)] who had at least one affected individual within one degree of separation, and 71 patients (29 male and 42 female with 37.7613.0 years) with sporadic PD	SNP’s in the *SNCA* gene are linked to an increased risk of developing PD
Kantor et al. [97]	Experimental gene therapy study	Human induced pluripotent stem cell (hiPSC)-derived dopaminergic neurons from a PD patient with the *SNCA* triplication	DNA hypermethylation *at SNCA* intron 1 allows an effective and sufficient tight downregulation of *SNCA* expression levels
Jankovic et al. [98]	Multicenter, randomized, double-blind, placebo-controlled, multiple ascending-dose trial	80 Caucassian PD patients; Effect of PRX002	PRX002 immunotherapy was capable of engaging peripheral α-synuclein in patients with PD.
Silveira et al. [99]	Single center, randomized, double-blind, placebo-controlled trial	75 individuals with mild to moderate PDD; effect of Ambroxol on GCase	Ambroxol could raise GCase and could therefore be a disease-modifying treatment for PDD
Alcalay et al. [100]	Multivariate genotyping analysis of *GBA* SNPs	517 PD patients and 252 controls with and without *GBA* mutations; *LRRK2*	Low glucocerebrosidase enzymatic activity may be a risk factor for Parkinson’s disease
Lesage et al. [101]	Genotyping analysis of *GBA* mutations	525 European (mostly French) PD patients from unrelated multiplex families, 605 patients with apparently sporadic PD and 391 ethnically matched controls	Higher incidence of L-dopa induced dyskinesias in *GBA*-PD patients
Zhang et al. [102]	Genotyping analysis + meta-analysis	1147 Chinese PD patients for L444P detection; Subsequent comparison between 646 PD patients with *GBA* mutations and 10344 PD patients without *GBA* mutations worldwide	Phenotypes of PD patients with *GBA* mutations are different from *GBA* non-carriers
Kasten et al. [103]	Systematic review	3652 citations; based on fully curated phenotypic and genotypic data on >1100 patients with recessively inherited PD because of 221 different disease-causing mutations in *Parkin, PINK1 or DJ1*	Mutations in the *PRKN* gene can lead to early onset PD, characterized by a clinically typical form of PD that is often associated with dystonia and dyskinesia
Khan et al. [104]	A phenotypic study of a large case series	24 patients with mutations in the *parkin* gene	Dyskinesias can occur early on in the course of the disease with very low doses of L-dopa

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
