# Peer review of "A Systematic Review of Parkinson’s Disease Pharmacogenomics: Is There Time for Translation into the Clinics?"

_ijms, 2021, doi:10.3390/ijms22137213_

Round 1
Reviewer 1 Report
the topic is interesting and the effort you have made considerable. The conclusions seem vague to me, however. The practical implication is weak and the work done not very usable .Author Response
Review comments
the topic is interesting and the effort you have made considerable. The conclusions seem vague to me, however. The practical implication is weak and the work done not very usable .
Response to Reviewer 1:
Dear collegue,
Thank you for the time you took to review our manuscript. We agree that our systematic review did not support the immediate translation of pharmacogenomics into clinical use for Parkinson's disease. Therefore, the conclusions might seem to be not »usable«.
However, we made this point very clear in the abstract and discussion. Furthermore, we firmly believe that »negative results« are very useful in identifying gaps for further research, as indicated in our paper. The importance of »negative results« is also demonstrated by Cochrane reports which often lack support for translation of research results into the clinics but are nevertheless highly valued by the clinical community.
We have made changes to the conclusions in the manuscript to make it clearer to the readers and present it now in a numbered list as follows:
„5. Conclusions
Most pharmacogenomic data for PD treatment present today is still not consistent enough to be entered into clinical practice, and further studies are required to enable a more personalized approach to therapy for each patient.The main findings can be summarized as follows:
- Most evidence from the analysed studies is found via secondary endpoints, which limits their power, with small sample size also being a diminishing factor.
- Conflicting reports between varied populations could be a consequence of low sample sizes and unaccounted interactions, which ultimately leads to low confidence in currently available data.
- The most promising avenues for clinical implementation of pharmacogenetics lie in the current findings of impulse control disorders and hyperhomocysteinemia, where the available data is more consistent.
- Most of the studies focus on L-dopa and DA, and greater focus should be also given to other PD treatment options such as MAO-B and COMT inhibitors in future studies.
Even though the wealth of knowledge is rapidly increasing, there is still not enough consistent data to make quality choices in the clinical treatment of patients. Studies that have a clear focus on pharmacogenomic properties of antiparkinsonian drugs are key for consolidating the current information and for the translation into clinical practice.“
Furthermore, the results are presented in a way that we have first covered the pharmacogenomic properties of major drug types used in treating Parkinson's disease, which is then followed by the specific pharmacogenomic properties of genotype driven Parkinson's disease. We are open to suggestions as to how we could improve the overall readability for the readers.

Reviewer 2 Report
In this systematic review, the authors have provided an assessment of the current evidence in the clinical translation of pharmacogenomics to personalize the treatment for Parkinson’s disease.
Greater attention was given to levodopa treatment and the COMT enzymatic pathway polymorphisms. This rationale is because L-dopa is commonly prescribed and is always combined with DDC inhibitors that alter L-dopa metabolism to the COMT pathway, increasing the bioavailability of L-dopa.
Should this restricted focus be reflected in the title?
The selection process is clearly indicated in figure 1. The discussion is balanced and covered all the relevant points related to the selected publications.
Author Response
Reviewer comments
In this systematic review, the authors have provided an assessment of the current evidence in the clinical translation of pharmacogenomics to personalize the treatment for Parkinson’s disease.
Greater attention was given to levodopa treatment and the COMT enzymatic pathway polymorphisms. This rationale is because L-dopa is commonly prescribed and is always combined with DDC inhibitors that alter L-dopa metabolism to the COMT pathway, increasing the bioavailability of L-dopa.
Should this restricted focus be reflected in the title?
The selection process is clearly indicated in figure 1. The discussion is balanced and covered all the relevant points related to the selected publications.
Response to reviewer
Dear collegue,
Thank you for the time you took to review this manuscript. The reviewer has well recognized concentration of available reports on the COMT enzymatic pathway. Nevertheless, this is simply the result of more papers published on this pathway and was not the consequence of this systematic review's» a priori«focus/hypothesis. We aimed to include all papers reporting on pharmacogenetics in Parkinson's. Stating the COMT pathway in the title might be misleading about the COMT pathway's impact, so we would rather keep the title as in the original manuscript. However, we have made some changes to the conclusion paragraph that addresses comments from both reviewers to make it clearer for the audience. The changed paragraph is as follows:
„5. Conclusions
Most pharmacogenomic data for PD treatment present today is still not consistent enough to be entered into clinical practice, and further studies are required to enable a more personalized approach to therapy for each patient.The main findings can be summarized as follows:
- Most evidence from the analysed studies is found via secondary endpoints, which limits their power, with small sample size also being a diminishing factor.
- Conflicting reports between varied populations could be a consequence of low sample sizes and unaccounted interactions, which ultimately leads to low confidence in currently available data.
- The most promising avenues for clinical implementation of pharmacogenetics lie in the current findings of impulse control disorders and hyperhomocysteinemia, where the available data is more consistent.
- Most of the studies focus on L-dopa and DA, and greater focus should be also given to other PD treatment options such as MAO-B and COMT inhibitors in future studies.
Even though the wealth of knowledge is rapidly increasing, there is still not enough consistent data to make quality choices in the clinical treatment of patients. Studies that have a clear focus on pharmacogenomic properties of antiparkinsonian drugs are key for consolidating the current information and for the translation into clinical practice.“
